# Rapid Detection of Malaria Based on Hairpin-Mediated Amplification and Lateral Flow Detection

**DOI:** 10.3390/mi14101917

**Published:** 2023-10-09

**Authors:** Yang Zhang, Lihui Ke, Tao Sun, Yang Liu, Bo Wei, Minghua Du

**Affiliations:** 1Comprehensive Technical Service Center of Xuzhou Customs, Xuzhou Customs, Xuzhou 221000, China; nuannuanyu@163.com; 2Department of Thoracic Surgery, Beijing Tiantan Hospital, Capital Medical University, Beijing 100070, China; k_ever96@163.com; 3Nanjing Customs, Nanjing 210001, China; 15365066618@163.com; 4Department of Health and Quarantine, Nanjing Customs, Nanjing 210001, China; njcoopy@163.com; 5Department of Emergency, The First Medical Center, Chinese PLA General Hospital, Beijing 100853, China

**Keywords:** hairpin-mediated amplification, lateral flow detection, Plasmodium

## Abstract

Malaria is listed as one of the three most hazardous infectious diseases worldwide. Travelers and migrants passing through exit and entry ports are important sources of malaria pandemics globally. Developing accurate and rapid detection technology for malaria is important. Here, a novel hairpin-mediated amplification (HMA) technique was proposed for the detection of four Plasmodium species, including *P. falciparum*, *P. vivax*, *P. malariae*, and *P. ovale*. Based on the conserved nucleotide sequence of Plasmodium, specific primers and probes were designed for the HMA process, and the amplicon can be detected using lateral flow detection (LFD); the results can be read visually without specialized equipment. The specificity of HMA-LFD was evaluated using nucleic acids extracted from four different Plasmodium species and two virus species. The sensitivity of HMA-LFD was valued using 10× serial dilutions of plasmid containing the template sequence. Moreover, 78 blood samples were collected to compare HMA-LFD and qPCR. The HMA-LFD results were all positive for four different Plasmodium species and negative for the other two virus species. The sensitivity of HMA-LFD was tested to be near five copies/μL. The analysis of clinical samples indicated that the consistency of HMA-LFD and qPCR was approximately 96.15%. Based on these results, the HMA-LFD assay was demonstrated to be a rapid, sensitive, and specific technique for the detection of Plasmodium and has great advantages for on-site detection in low-resource areas and exit and entry ports.

## 1. Introduction

Malaria is mainly endemic in tropical and subtropical regions, affecting more than 90 countries and areas worldwide while jeopardizing the health of 41% of people around the world; malaria has thus become a serious global public health issue [1]. Malaria has been listed by the World Health Organization as one of the three most hazardous infectious diseases to human health, in addition to the acquired immune deficiency syndrome (AIDS) and tuberculosis. Accurate and effective diagnostic methods are important for effectively controlling the spread of malaria. Developing accurate and rapid detection technology for malaria has been endorsed by the World Health Organization as one of the prioritized strategies in malaria management [2].

There are many techniques for detecting Plasmodium. Until recently, traditional thick and thin blood film microscopy was still the gold standard for malaria diagnosis [3,4,5], with the advantages of simplicity, low cost, and the ability to distinguish among different Plasmodium species. However, it has some limitations, such as the test results being easily affected by human factors, such as microscopic operation [6], low detection efficiency, and easy misdiagnosis and missed diagnosis [7]. Even though other techniques—including indirect fluorescence antibody tests [8,9,10], colloidal gold immunochromatography [11,12,13], and real-time polymerase chain reaction (qPCR) technology [14,15,16]—have addressed the challenge of rapid detection to varying degrees, they all have shortcomings, such as low sensitivity or low specificity [17,18]. Moreover, the processes involved in using these methods (including qPCR) are relatively complex and are reliant on the testing site, instruments and equipment, and technical personnel, making them not truly suitable for on-site rapid detection at exit and entry ports [19,20,21].

In this work, hairpin-mediated amplification (HMA) [22,23] and lateral flow detection (LFD) were utilized for the specific and efficient detection of nucleic acids from Plasmodium. HMA is a novel isothermal amplification technique for nucleic acids. Based on colloidal gold immunochromatography, its amplification products can be visually recognized on nucleic acid strips [22]. The temperature during HMA stays constant, which means that only a simple heater is required during the whole process. The HMA-LFD procedure is easy, and the results’ interpretation can be performed with only the naked eye. Moreover, both the amplification and detection processes are performed in an enclosed environment to prevent false-positive reactions arising from cross-contamination. These features will be conducive to a more effective application of the HMA-LFD technique in communities, facilitating the rapid detection of malaria at entry and exit port sites.

## 2. Experimental Methods

### 2.1. Materials and Reagents

A total of 78 anticoagulant blood samples were collected from Beijing Tiantan Hospital and Chinese PLA General Hospital. Informed consent was waived, given the characteristic of retrospective analysis in this work. All blood samples were taken via peripheral veins in 2% ethylenediaminetetraacetic acid (EDTA)-Na_2_ anticoagulant tubes and stored at −80 °C. Among these samples, 38 samples were microscopically determined to be positive for Plasmodium (including 23 cases of *P. falciparum*, 7 cases of *P. vivax*, 5 cases of *P. malariae*, and 3 cases of *P. ovale* positivity). The other 40 blood samples tested negative. All microscopic detections of Plasmodium were performed according to the guidelines of the WS 259-2006 Diagnostic Standards for Malaria [24].

A malaria real-time PCR kit was purchased from Shanghai Zhijiang Biotechnology, Shanghai, China. LFD and rapid nucleic acid extraction kits were purchased from Hangzhou Ustar Company, Hangzhou, China. The real-time PCR was performed using an ABI 7500, and ordinary PCR was performed using a PTC 200 (MJ Research Incorporated, San Francisco, CA, USA). A thermostatic metal bath was purchased from Thermo Fisher Scientific Inc., Coon Rapids, MN, USA. The microscope used was purchased from Nikon, Minato City, Japan.

### 2.2. HMA-LFD Protocol

**Rapid extraction of DNA templates**. A rapid nucleic acid extraction kit was used for DNA extraction. A total of 40 μL of rapid nucleic acid extraction reagent was added to 40 μL of whole blood sample. The mixture was placed at 95 °C for 5 min and then centrifuged at 10,000× *g* for 3 min. The supernatant was taken and placed at −20 °C for further use.

**Generation of plasmids containing target sequence.** In total, 2 μL of 10× buffer, 0.5 U of Taq DNA polymerase, 2 μL of extracted nucleic acid, and 2 μM of F3 and B3 were added to the reaction tube for PCR amplification. The process of PCR was set as pre-denaturation at 94 °C for 3 min, followed by a thermal cycle of denaturation at 94 °C for 20 s and annealing at 63 °C for 20 s, running for a total of 40 cycles. The amplification products were first validated by 2% agarose gel electrophoresis. The amplified sequences were combined with the pBS-T vector under the action of T4 DNA ligase, and transformed competent DH5a cells were then chosen for final plasmid construction. Next, positive colonies were sieved out for subsequent quantity enrichment on Lysogeny broth (LB) plates containing ampicillin. Plasmids were extracted from selected bacterial colonies, and the target fragments were verified to be successfully inserted into the plasmid by sequencing.

**Lateral flow detection.** The LFD strip contains the sample pad, conjugated pad, nitrocellulose membrane, and absorbent pad. The conjugated pad holds the matrix containing streptavidin conjugated red nanoparticles (30–50 μm), whereas the sample pad is utilized to load HMA amplicons. The end-point products’ ability to bind to the control and test lines, respectively, is made possible by the insertion of biotin and fluorescein isothiocyanate (FITC) at the 5′ end of the HMA primers [25,26].

**Design of HMA primer and probe.** Primers for HMA reactions were designed targeting the shared conserved sequences in the 18S rRNA genes of *P. falciparum*, *P. malariae*, *P. ovale,* and *P. vivax*. The respective primer sequences, including self-folding primers (SFP), turn back primers (TBP), artificial primers (AP), loop primers (LP), and two outer primers (F3/B3), are shown in Table 1. The integrated DNA technologies analysis tool (https://sg.idtdna.com/calc/analyzer (accessed on 20 December 2022)) assessed the self-folding circumstances of primers and primer–primer interactions to prevent the primer–primer interactions, which may result in false-positive amplification, especially for AP and LP. All the primers and probes were synthesized by Sangon Co., Ltd. (Shanghai, China).

**HMA amplification reaction and detection system.** The volume of the HMA reaction system was 20 μL, including 6 units of *Bst* 2.0 Warmstart DNA polymerase, 6 mM MgSO_4_, 10 mM dNTP, 0.1 μM F3/B3, 1.5 μM SFP, 1.0 μM TBP, 0.5 μM AP/LP, and 2 μL of the extracted template. The reaction tube was soaked in a metal bath for 40 min for whole HMA amplification. After amplification, 5 μL of the amplification product was dropped directly into the sampling area of the LFD, and then, 100 μL of 2× Saline Sodium Citrate (SSC) was added for detection. During the LFD detection processing, the 2× SSC functioned as a running buffer to take the HMA amplicon and nanoparticle from the sample pad to the absorbent pad, with the amplicon of HMA and the streptavidin-labeled nanoparticle being captured by the test line and control time, respectively [27].

For validated detection, both the quality control line and the detection line need to show colored bands, and when the color of the detection line band is deeper than the color of the colorimetric card strip, the result is judged to be positive; otherwise, it is negative. If neither the quality control line nor the detection line appears, the result is deemed invalid.

### 2.3. Performance Testing of HMA-LFD

**Specificity of HMA-LFD detection.** Whole blood samples containing four types of Plasmodium malaria parasites (*P. falciparum*, *P. malariae*, *P. ovale,* and *P. vivax*) and whole blood samples containing two other common pyrogenic pathogens (yellow fever virus and dengue virus) were prepared, and their respective nucleic acid templates were prepared according to the method described in Section 2.2. Then, six different samples were analyzed using the HMA-LFD technique to verify its specificity.

**Sensitivity of HMA-LFD detection.** The extracted plasmid DNA was quantified with a Nanodrop, and then, its concentration was calculated based on the fragment size of the plasmid. The quantified plasmid was then diluted by a 10× concentration gradient with 0.1× Tric-EDTA buffer. The diluted plasmid was used as a template for HMA and assayed accordingly with LFD to verify the sensitivity of HMA-LFD.

**Comparison of the detection efficiency of HMA-LFD and qPCR using clinical samples.** Rapid nucleic acid extraction was performed on 78 clinical whole blood samples using the rapid DNA extraction kit, and then, 4 μL of extracted samples was used as the template for HMA-LFD and qPCR detection. The criteria for determining the negative and positive qPCR results were carried out exactly in accordance with the product instructions.

## 3. Results

### 3.1. Optimization of the HMA-LFD System 

A DNA polymerase with strand displacement functionality and a set of specially created primers are required for the amplification of HMA [28], with the basic mechanism described in Figure 1A. The synthesized products primed from SFP and TBP are liberated from the template strand thanks to the strand displacement function of *Bst* DNA polymerase and the extension primed by outer primers. These dissociated single strand products, which feature an asymmetric dumb-bell structure, are useful for subsequent auto-cycling amplification stages. Due to intermolecular self-folding, the final amplicon of HMA can be changed into a conformation with an asymmetric dumb-bell shape, which results in the production of a following amplicon with a greater molecule weight [29,30]. Moreover, its amplification products can be visually recognized on nucleic acid strips based on colloidal gold immunochromatography.

The use of hairpin primers and the amplification time have a considerable impact on the detection efficiency in the HMA system; therefore, it is necessary to optimize the concentration of hairpin primers (Figure 1B) and the reaction time (Figure 1C). When the plasmid template was introduced into the amplification system at around 100 copies/μL concentrations, the detection sensitivity of HMA-LFD decreased as the TBP concentration decreased (from 1.5 μM to 0.5 μM). However, prolonging the reaction time of HMA (from 30 min to 40 min) can effectively improve the sensitivity of detection. The results were further confirmed by analyzing the gray intensity of the test line normalized to the control line (Figure 1D,E).

### 3.2. Specificity of HMA-LFD Detection System 

Samples containing four species of Plasmodium (*P. falciparum*, *P. vivax*, *P. ovale,* and *P. malariae*) or yellow fever virus and dengue virus nucleic acid were amplified by the HMA method and visualized by agarose gel electrophoresis, as well as LFD. The results are shown in Figure 2A and Figure 2B, respectively. Moreover, the results were also confirmed by extracting the gray values of the test line normalized to the control line (Figure 2C). The results showed that the samples containing nucleic acids from four different species of Plasmodium were all positive, while the samples containing the other two non-malaria pathogens were negative, proving that HMA-LFD could detect all different types of Plasmodia without the cross-identification of other viruses, indicating good specificity for malaria detection.

### 3.3. Sensitivity of HMA-LFD Detection System

The concentration of *P. falciparum* recombinant plasmid was measured as 5.34 × 10^8^ copy/μL, and the sample was diluted by 10 folds. Samples with theoretical concentrations of 5.34 × 10^4^ copy/μL, 5.34 × 10^3^ copy/μL, 5.34 × 10^2^ copy/μL, and 5.34 × 10^1^ copy/μL were used for analysis by qPCR and HMA-LFD systems. The results (Figure 3) showed that the two detection methods reached the same detection sensitivity of 5.34 × 10^1^ copy/μL.

### 3.4. Validation of the HMA-LFD Detection System Using Clinical Samples

A total of 78 whole blood samples from two clinical centers were analyzed with both qPCR and the HMA-LFD technique, as shown in Table 2. By using the results from the qPCR test as the standard, the detection sensitivity and specificity of the HMA-LFD method were 97.4% (37/38) and 95.0% (38/40), respectively, and the compliance rate of the two methods was 96.2% (75/78).

## 4. Discussion 

Malaria, as an infectious disease, seriously jeopardizes human health, and there are approximately 2 billion people at risk of malaria globally. Worldwide, approximately 250 million people are infected with malaria, and 1 million of them die every year [1]. At present, the gold standard for malaria detection is still thick and thin blood film microscopy, but this method is time consuming, laborious, greatly affected by the subjective experience of the examiner, and easily misses the diagnosis; thus, it cannot meet the needs of malaria examination and epidemic surveillance [31]. The rapid diagnosis method based on immunocolloidal gold technology has the problem of unstable detection results. PCR, real-time PCR, and gene chip technologies are capable of high-throughput detection with high sensitivity and specificity, but they are time consuming, expensive, relatively complex, and require expensive instruments [19]. Based on the above reasons, those detection methods are incapable of providing on-site rapid testing at entry and exit ports.

HMA technology is a new isothermal amplification technique. The HMA system consists of a pair of removable primers, cross-amplification primers, and hairpin primers, as well as an artificial sequence with biotin at the 3′ end and a target-sequence-specific probe with avidin labeling at the 3′ end [22,32]. The cross-primers include a target-specific hybridization sequence (3′ end) and an interchange sequence (5′ end), and the latter is the same as the 3′ end sequence of the hairpin primer. The artificial sequence coupled with biotin is consistent with the hairpin region in the hairpin primer. During amplification, both the cross-amplification primers and the corresponding removable primers can be attached to the template and lead to respective fragment extension. Under the action of *Bst* DNA polymerase, the extension products of the removable primers can remove those of the cross-amplified primers from the template [33,34]. The removed products generated by using the cross-amplified primers can be hybridized with hairpin primers and continue to extend accordingly to obtain a dumb-bell-like structure. The dumb-bell-like structure can mediate highly efficient amplification of artificial sequences and target-sequence-specific probes [35], resulting in double-stranded amplification products, which are simultaneously labeled with biotin and avidin, and the dual-labeled products can be detected using LFD to obtain a positive result. Correspondingly, the double-stranded amplification products of biotin and avidin, which are not labeled at the same time, can be tested as negative controls with LFD.

The results of this work show that the detection sensitivity of HMA-LFD is approximately five copies/μL based on the detection of different concentrations of the recombinant plasmid under gradient dilution, and this sensitivity is comparable to the sensitivity of the LAMP technique reported in previous literature [36,37]. HMA-LFD was used to detect the nucleic acids of three pathogens, which commonly cause febrile diseases in inbound and outbound personnel, and the results showed that only the nucleic acid of Plasmodium was positively detected, while the samples containing the other two pathogens remained negative. Therefore, HMA-LFD possessed good specificity. By using the qPCR test results as the gold standard, the testing results of 78 whole blood samples revealed that the consistency rate between the HMA-LFD technique and qPCR reached 96.2%, and the sensitivity and specificity were 97.4% and 95.0%, respectively.

In comparison with other nucleic acid amplification methods, the HMA-LFD technology has the following advantages: (1) The process has high reaction efficiency, as this work proved that the process of nucleic acid amplification can be completed within 40 min; (2) The reaction conditions for the HMA process are relatively less demanding and do not require professional personnel to achieve the whole testing procedure [38,39]. This study proves that the HMA-LFD technology is applicable for detecting Plasmodium. In addition to operating simplicity, its high sensitivity and specificity and significant consistency with qPCR contribute to the HMA-LFD technique being a suitable detection method for rapid malaria detection in people with suspected fever at entry and exit ports. However, the number of positive blood samples tested in the evaluation of the HMA-LFD technique in the current work was small, which may not fully and objectively reflect the effect of the HMA-LFD technique in the detection of malaria parasites. More positive samples will be tested in the future to more objectively and comprehensively assess the efficiency of HMA-LFD in Plasmodium detection. 

## 5. Conclusions

Malaria is a severe global public health concern, which is primarily prevalent in tropical and subtropical regions. Therefore, a quick and accurate diagnostic tool is essential to stop the disease from spreading. Here, we employed a cutting-edge isothermal amplification technique called HMA to quickly and accurately detect Plasmodium. Together with an artificial primer, the HMA primers identify five distinct sequences within the target sequence. After the self-folding of initial products, the creation of an asymmetric dumb-bell shape is what drives the amplification of HMA, which produces biotin/FITC co-labeled amplicons, which exhibit positive results utilizing LFD with high specificity. By using the qPCR test results as the gold standard, the testing results of 78 clinical samples revealed that the consistency rate between the HMA-LFD technique and qPCR reached 96.2%, and the sensitivity and specificity were 97.4% and 95.0%, respectively. For the future development and use of nucleic-acid-based POC diagnosis for resource-limited areas, HMA can be integrated with portable devices due to its high sensitivity and specificity.

## Figures and Tables

**Figure 1 micromachines-14-01917-f001:**
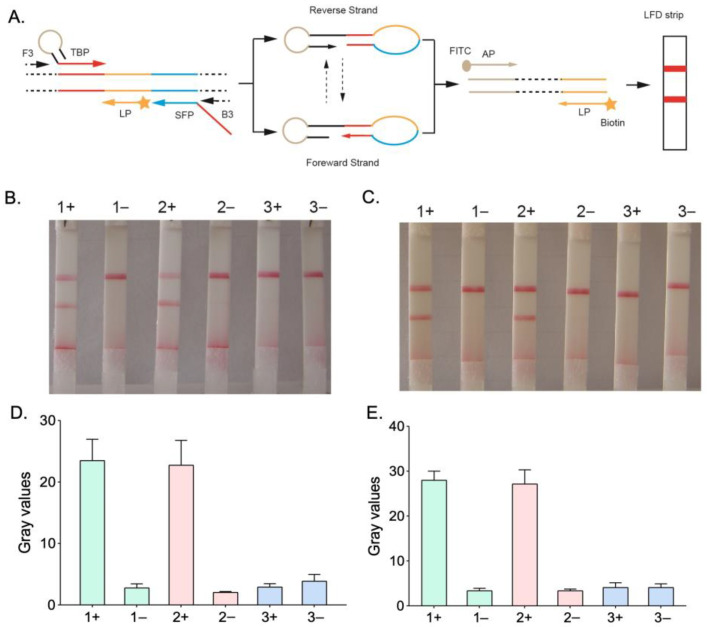
Brief mechanism and optimization of HMA-LFD reaction systems. (**A**) Brief mechanism of HMA amplification; (**B**) Effect of different TBP reaction concentrations on HMA-LFD detection sensitivity. The concentrations of TBP used were as follows: (1) 1.5 μM, (2) 1.0 μM, (3) 0.5 μM; (**C**) Effect of different reaction times on HMA-LFD detection sensitivity. The reaction times were (1) 40 min, (2) 35 min, and (3) 30 min. The templates were + (100 copies/μL plasmid template) and − (water as a negative control) samples; (**D**) Extracted gray values of optimization of the concentration of hairpin primers normalized to the controls; (**E**) Extracted gray values of optimization of the reaction times normalized to the controls.

**Figure 2 micromachines-14-01917-f002:**
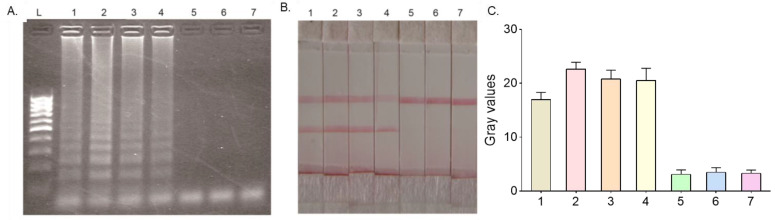
Specific testing of the HMA-LFD system for the detection of Plasmodium. The samples used included samples containing (1) *P. falciparum*; (2) *P. vivax*; (3) *P. ovale*; (4) *P. malariae*; (5) Dengue virus; (6) Yellow fever virus; (7) Blank controls. (**A**) 2% agarose gel electrophoresis test results; (**B**) HMA-LFD test results. (**C**) Extracted gray values.

**Figure 3 micromachines-14-01917-f003:**
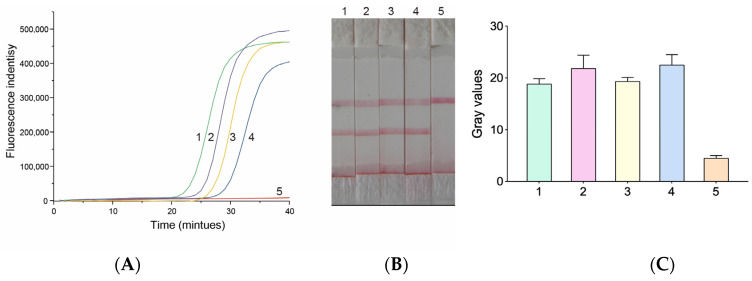
Detection sensitivity of qPCR and HMA-LFD. A plasmid containing a specific target sequence (*P. falciparum*) was used as a template, followed by a 10-fold concentration gradient dilution. (1) 10^4^ copy/μL; (2) 10^3^ copy/μL; (3) 10^2^ copy/μL; (4) 10^1^ copy/μL; (5) Negative control. (**A**) qPCR test results; (**B**) HMA-LFD test results; (**C**) Extracted gray values.

**Table 1 micromachines-14-01917-t001:** HMA-related primer sequence information.

Primer	Sequence (5′ to 3′)
F3	5-TCGCTTCTAACGGTGAAC
B3	5-AATTGATAGTATCAGCTATCCATA
SFP	5-GGTGGAACACATTGTTTCATTTGATCTCATTCCAATGGAACCT
TBP	5-TAACCACAGCCAGGTTAGGTGCTCGTGGTTAGGTGGAACACATTGTTTCATT
AP	5-FITC-TAACCACAGCCAGGTTAG
LP	5-BIOTIN-TGGACGTAACCTCCAGGC

**Table 2 micromachines-14-01917-t002:** Comparison of results of qPCR and HMA-LFD.

HMA-LFD	qPCR	
	Positive	Negative
Positive	37	2
Negative	1	38

## Data Availability

Not applicable.

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
