# Peer review of "Rapid Detection of Malaria Based on Hairpin-Mediated Amplification and Lateral Flow Detection"

_micromachines, 2023, doi:10.3390/mi14101917_

Round 1

Reviewer 1 Report

Dear authors,

I have carefully reviewed your manuscript and have several recommendations to improve the overall quality and clarity of the article before it can be considered for publication:

1) Thorough Experimental Section: The experimental section needs more detail. Please provide a comprehensive description of the experimental design, including information on primer design, methods, preparation of the lateral flow, membrane porosity, detection line preparation, buffers used, incubation times, and whether any conjugation pad, wicking pad, or cassette was employed. Additionally, please include proper references for the materials used.

2) Explain Hairpin-Mediated Amplification: The mechanism of hairpin-mediated amplification (HMA) is mentioned but not adequately explained in the manuscript (only in the Conclusions section). It should be detailed in the introduction to avoid reader confusion.

3) Include Schematics: It would greatly enhance clarity if you include schematic representations of how HMA works, illustrating the role of removable primers, cross-amplification primers, and hairpin primers. Additionally, provide a schematic on how the detection process is carried out.

4) Clarify Sequences in Table 1: The sequences listed in Table 1 are referred to as "primers," but their specific roles are not explained. Please provide clarity on the function of each sequence.

5) Specify Gold Particles: Mention the type of gold particles used in the colloidal gold immunochromatography-based detection mechanism. Include information on any modifications, conjugations, and part numbers. Ensure this is covered in the experimental section.

6) Remove Unused FITC Sequence: Remove the sequence with FITC from Table 1 if it is not used in the experimental section or elsewhere in the manuscript.

7) Define Acronyms: The manuscript contains numerous acronyms that are not explained. Define acronyms such as "pBS-T," "PGF3," "PGB3," "LB," "PGCPF," "PGCPR," "PGDR5F," "PGDR5B," "2XSSC," and others for clarity.

8) Fig 1.A and Fig 1.B Clarifications: In Fig 1.A, specify the concentration of the target in terms of copy number, not molarity. Additionally, please provide the concentration of PGCPR in Fig 1.B.

9) Reorganize Results and Conclusion: Currently, some discussion of results is found in the "Conclusions" section. Consider moving this discussion to a more appropriate section and rework the "Conclusions" section accordingly.

Addressing these points will significantly enhance the quality and clarity of your manuscript, making it more suitable for publication. Thank you for your attention to these recommendations.

Author Response

Thank you very much for taking the time to review this manuscript. Please find the detailed responses below and the corresponding revisions/corrections highlighted in the re-submitted files.

Reviewer 2 Report

The HMA-LFD technique offers a promising solution for the rapid and accurate detection of multiple Plasmodium species, crucial in combating malaria. Its simplicity, sensitivity, and compatibility with low-resource settings and entry/exit points make it a valuable tool for malaria control and prevention efforts.

The objective holds significance and should engage Micromachines' readers. Nevertheless, this manuscript displays certain structural weaknesses. Hence, I recommend a thorough reorganization before resubmission to Micromachines or other journals.

1.      The inclusion of a schematic illustrating the system's operation is essential as the current description is challenging to comprehend.

2.      The 'Conclusions' section should be merged into the 'Discussion,' and a new 'Conclusions' section must be added.

3.      Figures 1 to 3 should provide quantitative data rather than binary results (yes or no) for lateral flow detection, possibly employing image analysis tools.

Overall, the topic's importance is undeniable, but improvements in organization and data analysis are required to meet Micromachines' article standards. I encourage the author to revise the manuscript and consider submission to alternative journals.

Author Response

(The authors gave the same response as above.)

Reviewer 3 Report

The manuscript  by Zhang et al present the hairpin-mediated amplification and lateral flow detection device for malaria detection. The article is well writen, the introduction is comprehensible and the figures make a well strucured manuscript. I only have the small remarks below:

1. What concentration was used to test the reaction time?

2. Developing accurate and rapid detection technology for maleria detcetion for travelers and migrants passing through exit and entry ports is the main goal, what is the expected price for the final product?

Line 61 change ”judgement of the results” with  “the results interpretation”

Author Response

(The authors gave the same response as above.)

Round 2

Reviewer 1 Report

I believe the authors have addressed the previous comments in a satisfactory way and the manuscript can be now considered for publication

Reviewer 2 Report

The author has responded to all my questions. I am happy to see it published in the current version.